# Infective Endocarditis in Belgium: Prospective Data in Adults from the ESC EORP European Endocarditis Registry

**DOI:** 10.3390/jcm13051371

**Published:** 2024-02-28

**Authors:** Bram Roosens, Bernard Cosyns, Patrizio Lancellotti, Cécile Laroche, Christine Selton-Suty, Agnès Pasquet, Johan De Sutter, Philippe Unger, Bernard Paelinck, Paul Vermeersch, Andreea Motoc, Xavier Galloo, Bernard Iung, Gilbert Habib

**Affiliations:** 1Centrum voor Hart-en Vaatziekten (CHVZ), Vrije Universiteit Brussel (VUB), Universitair Ziekenhuis Brussel (UZ Brussel), 1090 Brussels, Belgiumphilippe.unger@ulb.be (P.U.); bernard.paelinck@uza.be (B.P.); paul.vermeersch@zna.be (P.V.); xavier.galloo@uzbrussel.be (X.G.); 2Department of Cardiology, Centre Hospitalier Universitaire Sart Tilman, 4000 Liège, Belgium; plancellotti@chu.ulg.ac.be; 3EURObservational Research Programme, European Society of Cardiology, 06903 Biot, Francec.suty-selton@chru-nancy.fr (C.S.-S.); 4Department of Cardiology, Centre Hospitalier Universitaire de Nancy, 54000 Nancy, France; 5Divisions of Cardiology and Cardiothoracic Surgery, Cliniques Universitaires Saint-Luc, 1200 Brussels, Belgium; agnes.pasquet@saintluc.uclouvain.be; 6Hartcentrum Gent, Algemeen Ziekenhuis Maria Middelares, 9000 Ghent, Belgium; 7Department of Cardiology, Centre Hospitalier Universitaire Saint-Pierre, 1000 Brussels, Belgium; 8Department of Cardiology, Universitair Ziekenhuis (UZ) Antwerpen, 2650 Antwerp, Belgium; 9Hartcentrum, Ziekenhuis Netwerk Antwerpen (ZNA), 2020 Antwerp, Belgium; 10Department of Cardiology, Bichat Hospital, APHP, Université Paris-Cité, 75018 Paris, France; bernard.iung@aphp.fr; 11APHM, La Timone Hospital, Cardiology Department, Marseille, France & Aix Marseille University, IRD, APHM, MEPHI, IHU-Mediterranean Infection, 13005 Marseille, France

**Keywords:** Belgium, cardiac surgery, infective endocarditis, registry, valve disease

## Abstract

(1) Background: infective endocarditis (IE) is a significant health concern associated with important morbidity and mortality. Only limited, often monocentric, retrospective data on IE in Belgium are available. This prospective study sought to assess the clinical characteristics and outcomes of Belgian IE patients in the ESC EORP European endocarditis (EURO-ENDO) registry; (2) Methods: 132 IE patients were identified based on the ESC 2015 criteria and included in six tertiary hospitals in Belgium; (3) Results: The average Belgian IE patient was male and 62.8 ± 14.9 years old. The native valve was most affected (56.8%), but prosthetic/repaired valves (34.1%) and intracardiac device-related (5.3%) IE are increasing. The most frequently identified microorganisms were *S. aureus* (37.2%), enterococci (15.5%), and *S. viridans* (15.5%). The most frequent complications were acute renal failure (36.2%) and embolic events (23.6%). Cardiac surgery was effectively performed when indicated in 71.7% of the cases. In-hospital mortality occurred in 15.7% of patients. Predictors of mortality in the multivariate analysis were *S. aureus* (HR = 2.99 [1.07–8.33], *p* = 0.036) and unperformed cardiac surgery when indicated (HR = 19.54 [1.91–200.17], *p* = 0.012). (4) Conclusion: This prospective EURO-ENDO ancillary analysis provides valuable contemporary insights into the profile, treatment, and clinical outcomes of IE patients in Belgium.

## 1. Introduction

Despite advancements in infective endocarditis (IE) management, it remains a significant health concern associated with important morbidity and mortality [1,2]. Annually, there are an estimated 13.8 IE cases per 100,000 individuals, resulting in 66,300 global deaths attributed to IE [2,3]. In the past decades, the epidemiologic profile of IE has considerably evolved, with alterations in affected populations, risk factors, and causative microorganisms. Moreover, developments in imaging modalities have been transformative in IE diagnostics [4]. Recently, ESC guidelines on the management of IE have been updated to reflect these insights [2]. Currently, only limited, often monocentric, retrospective data about IE in Belgium are available, and contemporary national epidemiological data are lacking [5,6,7,8]. The European Society of Cardiology (ESC) EURObservational Registry Programme (EORP) European Endocarditis registry (EURO-ENDO) is a multicentre, prospective, observational cohort study investigating the care and outcomes of IE patients at hospitals in Europe and ESC-affiliated/non-affiliated countries [9]. The EORP has been established to enhance the understanding of medical practice through observational studies with rigorous methodological procedures. This ancillary, prospective EORP sub-analysis of IE in Belgium aimed to provide an overview of the contemporary clinical characteristics, microbiological profile, diagnostic workup, in-hospital complications, surgical intervention, and mortality in a nation at the crossroads of Europe.

## 2. Materials and Methods

### 2.1. Study Design and Data Collection

The methodology of the ESC EORP EURO-ENDO registry has been previously reported [9]. From 1 January 2016 to 31 March 2018, 3116 IE patients aged at least 18 years were included from 156 centres across 40 countries. Follow-up over a one-year period lasted until April 2019. Centers taking part in the study were categorized into high-level IE centres, characterized by a substantial patient load (≥20 patients per year) and expertise in IE diagnosis, management, imaging, and surgical therapy; or low-volume centres (<20 patients per year), without surgical facilities. In Belgium, six tertiary high-volume hospitals actively participated in the registry: Algemeen Ziekenhuis (AZ) Maria Middelares Gent, Centre Hospitalier Universitaire (CHU) de Liège, CHU Saint-Pierre, Cliniques universitaires Saint-Luc, Universitair Ziekenhuis (UZ) Antwerpen and UZ Brussel. Potential participants were identified from both echocardiographic laboratories and hospitals that cater to patients with IE. Inclusion criteria were a diagnosis of definite IE (or possible IE, but considered and treated as IE) based on the ESC 2015 IE guidelines [4]. Definitions of the terms used in the ESC 2015 modified criteria for the diagnosis of IE are provided in Appendix A. IE was considered definite if any of the following criteria were met: two major clinical criteria, one major clinical criterion along with three minor criteria, or five minor criteria. A diagnosis of possible IE was considered when one major and one minor criterion were present or when three minor criteria were present. The study exclusion criteria were a patient age of less than 18 years and participation in other interventional clinical studies that would interfere with the patient’s usual care. Data collected at inclusion and during hospitalization comprised demographics (age, sex) and patient history (cardiovascular and other co-morbidities), clinical characteristics at presentation (signs and symptoms), previous invasive intervention within the last 6 months (dental, colonoscopy), microbiology (blood cultures, serology), imaging and its results (echocardiography, cardiac multislice Computed Tomography (CT), 18F-FDG positron emission tomography/computed tomography (18F-FDG PET/CT), cardiac magnetic resonance imaging (MRI)), complications (embolic events, haemodynamic), surgery, and in-hospital mortality. Data were pseudonymized using a unique patient study code [9]. This study complied with the Declaration of Helsinki. The locally appointed ethics committee approved the research protocol, and informed consent was obtained from all subjects (or their legally authorized representative).

### 2.2. Data Management and Statistical Analysis

Gathering officers at the participating sites were responsible for data collection, which was entered into a comprehensive online electronic case report form (CRF) encompassing 430 distinct data fields. ESC EORP management teams oversaw data quality. The first author, having full access to all study data, assumed responsibility for its integrity and the current data analysis. Continuous variables are presented as mean ± standard deviation. For group comparisons, the Kruskal–Wallis H test for non-parametric data was used. Categorical variables are expressed as frequency and percentages, with 2 × 2 among-group comparisons conducted using Pearson’s chi-squared χ^2^ test or Fisher’s exact test when expected cell counts were <5. In other instances, the Monte Carlo estimate of the exact *p*-value was applied. Univariable analysis was performed for both continuous and categorical variables. Prior to advancing to the backward multivariable Cox regression analysis to identify predictors of mortality, pairwise correlations between all candidate variables (with *p* < 0.10 in univariable analysis) were tested. Survival and event-free survival were also assessed using Kaplan–Meier curves. Several measures of the model of fit, including concordance and the goodness of fit test proposed by May and Hosmer, were considered. Proportional hazard ratio assumptions were graphically verified using the Schoenfeld residuals test. A value of *p* < 0.050 was considered significant. All analyses were performed using IBM SPSS Statistics software version 29.0.1.0 (171) (Chicago, IL, USA).

## 3. Results

132 Belgian IE patients were prospectively included. IE was definite in 120/132 (90.9%) patients and possible in 12/132 (9.1%) patients. 51/132 (38.6%) of IE patients were referred from regional centres to one of the participating, tertiary hospitals.

### 3.1. Patient Demographics and Characteristics

The demographics and characteristics of the patients are displayed in Table 1. Among the 132 IE patients, 47/132 (35.6%) underwent a previous heart valve intervention and 16/132 (12.1%) underwent a preceding intracardiac device implantation. Seventy-five (56.8%) patients were diagnosed with native valve IE (NVE, group 1), 45 (34.1%) with prosthetic or repaired valve IE (PRVE, group 2), and 7 (5.3%) with intracardiac device-related IE (CDRIE, group 3). The remaining five patients, with IE corresponding either to another or an undetermined location, were not included in the group comparison analysis. The valvular IE location was aortic in 76 (57.6%), mitral in 55 (41.7%), tricuspid in 6 (4.5%), and pulmonary in 3 (2.3%) IE patients. IE affected two or more valvular locations in 14.2% of the patients. There were no transcatheter aortic valve replacement (TAVR) or transcatheter mitral valve procedure (TMVP) IE cases in this study.

### 3.2. Clinical and Biological Features

The clinical features can be found in Appendix A. Fever (70.1%), cardiac murmur (49.6%), and embolic events (26.0%) were the most frequent symptoms at initial presentation.

Blood cultures were positive in 90% of Belgian IE patients. The microbiological results are shown in Appendix A. The most frequently identified microorganisms were *S. aureus* (37.2%), Enterococci (15.5%) and *S. viridans* (15.5%). Gram-negative HACEK infections were rare (2.7%). There were no *C. burnetii* infections and no significant differences in microbiological profiles between the groups.

### 3.3. Imaging

Echocardiography was performed at least once in 100.0%, transthoracic echocardiography (TTE) in 81.9%, and transoesophageal echocardiography (TOE) in 94.5% of Belgian IE patients. There were no significant differences in usage between the groups, nor in the echocardiographically assessed vegetation size or local IE complications.

18F-FDG PET/CT was performed in 46/127 (36.2%) and positive in 28/46 (60.1%) IE patients. There was 75% cardiac and 53.6% extra-cardiac uptake, without significant differences in utilization or findings between the groups. Cardiac CT (5.5%), leukocyte scintigraphy (3.1%) and cardiac MRI (0.0%) were rarely used, likewise without statistically significant differences between groups. Extra-cardiac CT was performed in 63.0% and revealed extra-cardiac lesions in 46.3% of IE patients.

### 3.4. In-Hospital Complications under Treatment

The main in-hospital complications are shown in Appendix A. Acute renal failure (36.2%) was the most frequent in-hospital complication in Belgian IE patients, followed by embolic events (23.6%, of which 46.7% were cerebral embolic events) and positive blood cultures after 48 h (20.5%). There were no significant differences between the groups.

### 3.5. Cardiac Surgery and Mortality

Following the ESC guidelines [4], the theoretical indication for cardiac surgery (103/127, 81.1%) was not significantly different between the groups (*p* = 0.092). The most frequent indication in all groups was infection (67.0%, *p* = 0.956). Surgery was effectively performed when indicated in 91/127 (71.7%) patients: 70.1% in NVE, 75.5% in PRVE, and 57.1% in CDRIE (*p* = 0.722). The most common reason for not performing surgery despite a theoretical indication was a high surgical risk (83.3%, *p* = 1.000), followed by death before surgery (8.3%) and neurological complications (8.3%). When operated on, bioprostheses were most often implanted (aortic valve position: 42.4% vs. 22.0% mechanical; mitral valve position: 47.1% vs. 26.5% mechanical). Mitral valve repair was only performed in 23.5% of patients with mitral valve IE.

In-hospital death occurred in 20/127 (15.7%) IE patients. The causes of mortality are shown in Table 2. Independent predictors of mortality by multivariable analysis in IE patients are shown in Table 3 and by univariate analysis in Appendix A.

Kaplan-Meier survival curves are provided for all-cause mortality according to group (Figure 1) and according to the performance of surgery when indicated (Figure 2).

After follow-up, an additional four patients died, resulting in a total one-year mortality rate of 18.9%.

## 4. Discussion

This is the first prospective, multicentric study of IE in Belgium. The following findings arise from this EURO-ENDO ancillary analysis: first, the average Belgian IE patient was male and approximately 63 years old. Second, the native aortic valve is most commonly affected, but PRVE and CDRIE are markedly increasing. Third, cancer is a frequent comorbidity in Belgian IE patients. Fourth, the most frequently identified microorganism was *S. aureus*, followed by *S. viridans* and enterococci. Fifth, when indicated, surgery was often performed. Mainly bioprostheses are used. Sixth, notwithstanding, IE remains a deadly disease.

### 4.1. Demographics, Clinical and Microbiological Characteristics

With a mean age of 62.8 ± 14.9 years, Belgian IE patients are notably older than the global IE population in EURO-ENDO (around 59 years old) [10]. This finding is in accordance with previous retrospective Belgian IE studies [5,8], but in stark contrast to the more commonly affected younger age groups outside of Europe (between 35 and 55 years old). This underscores regional variations in the epidemiology of this disease, with varying predisposing conditions, such as rheumatic heart disease and injection drug use [11,12]. Nevertheless, the mean age of IE patients has been gradually increasing globally since the 1990s [13,14], presenting treatment challenges due to additional comorbidities and frailty. This study further confirms a previously asserted 2:1 male-to-female ratio in IE [2].

The frequency of PRVE (34.1%) and CDRIE (5.3%) in Belgium is comparable to the global IE population, and the proportion of valvular IE locations is equally similar, with the aortic (57.6%) and mitral valves (41.7%) being most commonly affected [10]. In many European countries, the share of PVRE is steadily increasing since the 2000s due to older age [15]. In contrast, in countries with a younger IE population due to injection drug use, NVE remains much more common (90%), and mitral and tricuspid valve infections often exceed aortic valve IE [11,12].

Belgian IE patients commonly retain a history of arterial hypertension (53.5%), atrial fibrillation (29.9%), cancer (23.6%), and ischemic heart disease (19.7%), particularly in the slightly older PRVE and CDRIE groups. A previous EURO-ENDO ancillary study showed a cancer prevalence of 11.6% in the global IE population [16]. In Belgium, combined cardiovascular disease and cancer cause > 50% of all deaths, emphasizing the importance of investigating the shared pathophysiological mechanisms [17]. Cancer patients may be at an increased risk of developing IE, mainly after invasive procedures or due to compromised immunity [16]. In Belgian IE patients, immunosuppressive treatments (3.9%) and colonoscopy procedures (5.5%) were comparable to those in the global IE population [10]. The current study’s IE patients had distinctly less heart failure compared to the global EURO-ENDO IE population (11.8% vs. 23.3%), and less frequently presented with congestion (18.1% vs. 27.2%) and cardiogenic shock (1.6% vs. 2.3%) [10], possibly due to a referral bias, as patients with severe heart failure might have been deemed unfit to be transferred to a tertiary cardiac surgery centre.

There were only 10% blood culture-negative Belgian IE cases, compared to 20% in the global EURO-ENDO population [10], possibly testifying to improved diagnostic strategies, such as culturing on specialized media, systematic serological testing, polymerase chain reaction assays, and ribosomal ribonucleic acid sequencing [2]. A recent EURO-ENDO ancillary study showed increased mortality in blood culture-negative IE [18], but there was no significant difference in mortality between blood culture-positive and-negative IE patients in the current study (*p* = 0.307). The most frequently identified microorganism remains *S. aureus* (37.2%), which is a persisting concern given its potential for severe complications and high mortality rate [19]. On the other hand, the prevalence of enterococci (15.5%) has become equal to that of *S. viridans* IE in Belgium (15.5%), while these former infections were rarer in the past (around 10%) [14,15], but have been steadily increasing in Belgium since the 2000s [5,8]. This might be explained by the aging population and a higher cancer prevalence [20,21]. In contrast, outside Europe, *S. aureus* remains much more prevalent (45%), compared to streptococci (7%) and enterococci (6%) [11,12]. Together with *S. gallolyticus*, intestinal bacteria accounted for 23.7% of IE cases in this study. This microbiological shift was also observed in the global EURO-ENDO data, where Enterococcus overtook *S. viridans* IE (15.8% vs. 12.4%) [10]. As such, opportunities for IE prevention may arise during invasive urogenital and gastrointestinal procedures, particularly in older individuals or those with a cancer history [2,16,20].

### 4.2. Imaging

The transformative application of imaging techniques shortly after the publication of the 2015 ESC guidelines was similarly observed in Belgian IE patients, as TEE was performed in 94.5% (vs. 58.1% in the EURO-ENDO global population) and 18F-FDG PET/CT in 36.2% (vs. 16.6%) [4,10]. 18F-FDG PET/CT has proven useful in the diagnosis of PRVE and CDRIE [2]. However, in Belgian IE patients, there was only a marginally higher application in CDRIE (57.1%) and PRVE (40.0%) compared to NVE (32.0%, *p* = 0.463). In NVE, 18F-FDG PET/CT mainly has value in detecting peripheral lesions or adding minor diagnostic criteria [2]. In Belgium, there might still be room for improvement in the application of leukocyte scintigraphy (3.1%), cardiac MRI (0.0%), and especially cardiac CT (5.5%) for the detection of perivalvular infection in PRVE [2]. However, given the elapsed time since the EURO-ENDO registry and the currently more prevalent availability of these modern imaging methods, a more contemporary study might already show increased utilization.

### 4.3. Management and Outcome of IE in Belgium

#### 4.3.1. In-Hospital Complications

Acute renal failure is the most common complication and is remarkably more frequent than in the global EURO-ENDO population (36.2% vs. 17.7%) [10], possibly due to the older Belgian IE population with regularly underlying chronic renal failure (17.3%). In contrast, Belgian IE patients developed less congestive heart failure during hospitalisation (8.7%), possibly due to a previously mentioned referral bias. At admission, embolic events were already present in 26.0% of IE patients, and despite the initiation of antibiotic therapy, additional (especially cerebral) embolism was detected in 23.6%. These are frequent and life-threatening complications, with a high global incidence ranging from 13 to 49% in IE [5,19]. However, in this study (cerebral) embolism accounted for only 5% of all-cause mortality. In comparison, outside of Europe, pulmonary embolism is more frequently seen (up to 35%) and is typically related to right-sided IE due to injection drug use [22].

#### 4.3.2. Surgery

Surgery was performed in 71.7% of IE patients, which is high compared to the global IE-related interventions of about 50% since the 2000s [10,23] and a sometimes low cardiac surgery rate (13%) outside Europe [11]. Improved identification of patients eligible for surgery may have contributed to this number, as well as a referral bias to tertiary centres with cardiac surgery [24]. The theoretical indication for surgery following the ESC 2015 guidelines in Belgian IE patients was high (81.1%), and when indicated, cardiac surgery was often effectively performed. When denied surgery, the main reason was a high surgical risk, possibly related to old age, frailty, and comorbidities. Bioprosthetic valves were most often used in Belgian IE patients, more frequently than observed in the Euro heart survey of 2003, when mechanical prostheses were still more prominent (74%) [25]. This change might be related to older age and the possible need for further surgical procedures with a risk of bleeding under anticoagulant therapy [26]. Mitral valve repair is underused in Belgian (23.5%), as in global IE patients, perhaps due to excessive valvular destruction [10]. In contrast, several studies have shown that valve repair, when feasible, is associated with better outcomes [13,27].

#### 4.3.3. In-Hospital Mortality

In-hospital all-cause mortality was comparable to that of the global EURO-ENDO population (15.7% vs. 17.1%) and remains substantial [10]. There was a trend towards higher all-cause death in the CDRIE and PRVE groups (*p* = 0.051). The lower mortality compared to previous registries might once again be due to a referral bias to tertiary centres. The main driver of all-cause mortality in Belgian patients was non-cardiovascular disease, particularly sepsis (60.0%). By univariate Cox regression analysis, predictors of mortality were the Charlson index (HR = 1.18 [1.04–1.34], *p* = 0.011), creatinine > 2 mg/dL (HR = 3.49 [1.34–9.08], *p* = 0.011), and unperformed cardiac surgery when indicated (HR = 13.72 [1.56–120.61], *p* = 0.018). Given the significant impact of cancer on the Charlson index, it might be worthwhile to conduct a sub-analysis using a modified Charlson score that excludes cancer [16,28]. By multivariate analysis, adjusted for age, *S. aureus* (HR = 2.99 [1.07–8.33], *p* = 0.036) and especially unperformed cardiac surgery when indicated (HR = 19.54 [1.91–200.17], *p* = 0.012) remained significant, independent predictors of outcome. This further emphasizes the need for early discussion with surgeons within the IE team, as recently reemphasized by the 2023 ESC guidelines [2].

### 4.4. Study Limitations

This ancillary study has the same limitations as the EURO-ENDO registry, especially selection and referral bias, as all patients were recruited in high-level centres and inclusion was based on voluntary participation [10]. Therefore, estimating the true incidence of IE in Belgium is unfeasible based on the current dataset, patient characteristics might have been influenced, and the effect on mortality is unclear [24]. Additionally, while the present dataset dating back to 2016–2018 does not include TAVR or TMVP IE, these would be more frequently present in contemporary data collection and warrant future exploration. Furthermore, comprehensive data concerning the pivotal role of the IE team, highlighted in the latest guidelines [2], were missing in the current study. Future studies should elucidate its impact on clinical outcomes. The reasons for denial of surgery should also be investigated in future studies of IE patients, as well as the development of scores to predict the futility of surgical therapy [2]. This IE study had a restricted follow-up period of one year. However, given potential late relapse, complications, and mortality [8], future investigations should incorporate an extended follow-up. Finally, this was an ancillary analysis of a registry that was not dedicated to the Belgian population. The limited number of Belgian IE patients in the EURO-ENDO underscores the necessity for a future prospective, observational cohort study of IE in Belgium, proportionally including non-tertiary regional centers and with an extended inclusion period to improve recruitment and increase the study’s power. This might provide a more comprehensive understanding of this rare but deadly disease. These limitations were counterbalanced by the active participation of many tertiary centers and the quality of CRF completion in EURO-ENDO [9].

## 5. Conclusions

This prospective EURO-ENDO ancillary analysis provides valuable contemporary insights into the profile, diagnostics, treatment, and clinical outcomes of IE patients in Belgium, in light of recent updates in IE management recommendations. Belgian IE patients are older and often have cancer. The native aortic valve is frequently affected; however, PRVE and CDRIE are markedly increasing. *S. aureus*, followed by *S. viridans* and Enterococci are the primary micro-organisms. When indicated, surgery is often performed. Multivariate analysis revealed that *S. aureus* and unperformed cardiac surgery, when indicated, were independent predictors of outcome. Despite important study limitations, these data could nevertheless guide future research endeavors to enhance the management and prognosis of IE in Belgian healthcare.

## Figures and Tables

**Figure 1 jcm-13-01371-f001:**
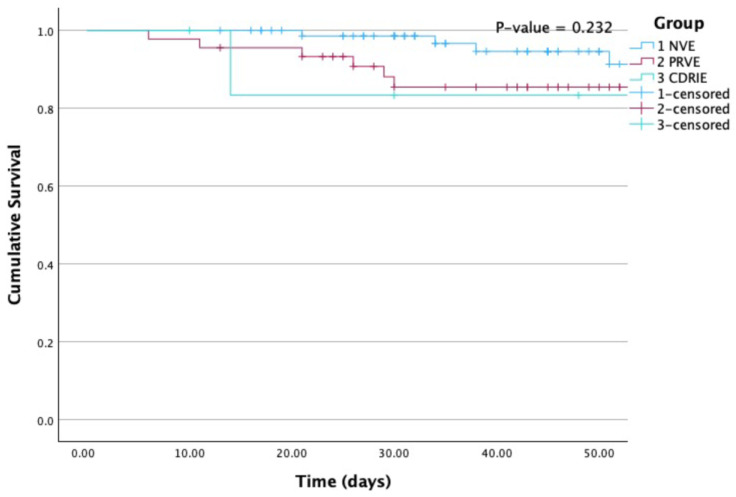
Kaplan-Meier curves for all-cause mortality according to group. NVE, native valve infective endocarditis; PRVE, prosthetic + repaired valve infective endocarditis; CDRIE, cardiac device-related infective endocarditis. *P* (Log Rank) = 0.232.

**Figure 2 jcm-13-01371-f002:**
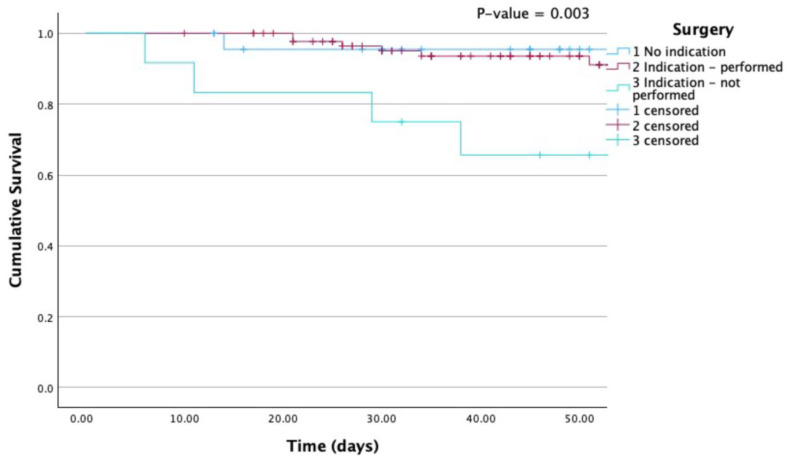
Kaplan-Meier curves for all-cause mortality according to surgery. Mortality was particularly elevated when surgery was indicated but not performed. *P* (Log Rank) 0.003.

**Table 1 jcm-13-01371-t001:** Demographics and clinical characteristics of Belgian infective endocarditis patients.

	Total	NVE	PRVE	CDRIE	*p*-Value
N	127	75	45	7	
Demographics					
Age (years)					
Mean ± SD	62.8 ± 14.9	60.6 ± 15.4	66.5 ± 14.0	63.6 ± 10.5	0.073
<65 years old	57/127 (44.9%)	40/75 (53.3%)	15/45 (33.3%)	2/7 (28.6%)	
65–80 years old	56/127 (45.1%)	29/75 (38.7%)	22/45 (48.9%)	5/7 (71.4%)	
≥80 years old	14/127 (11.0%)	6/75 (8.0%)	8/45 (17.8%)	0/7 (0.0%)	
Males (%)	92/127 (72.4%)	53/75 (70.1%)	33/45 (73.3%)	6/7 (85.7%)	0.801
History of cardiovascular diseases					
Atrial fibrillation	38/127 (29.9%)	15/75 (20.0%)	19/45 (42.2%)	4/7 (57.1%)	0.009 *
Known valve murmur	26/127 (20.5%)	12/75 (16.0%)	14/45 (31.1%)	0/7 (0.0%)	0.073
Ischemic heart disease	25/127 (19.7%)	6/75 (8%)	15/45 (33.3%)	4/7 (57.1%)	<0.001 *
Congenital heart disease	19/127 (15.0%)	10/75 (13.3%)	9/45 (20.0%)	0/7 (0.0%)	0.409
Heart failure	15/127 (11.8%)	5/75 (6.7%)	6/45 (13.3%)	4/7 (57.1%)	0.003 *
Previous IE (%)	7/127 (5.5%)	1/75 (1.3%)	5/45 (11.1%)	1/7 (14.3%)	0.024 *
Hypertrophic cardiomyopathy	4/127 (3.1%)	2/75 (2.7%)	2/45 (4.4%)	0/7 (0.0%)	0.703
Risk factors					
Arterial hypertension	68/127 (53.5%)	37/75 (49.3%)	27/45 (60.0%)	4/7 (57.1%)	0.531
Smoking	34/127 (26.8%)	25/75 (33.3%)	7/45 (15.6%)	2/7 (28.6%)	0.107
Cancer	30/127 (23.6%)	16/75 (21.3%)	14/45 (31.1%)	0/7 (0.0%)	0.176
COPD/asthma	23/127 (18.1%)	11/75 (14.7%)	8/45 (17.8%)	4/7 (57.1%)	0.034 *
Chronic renal failure	22/127 (17.3%)	9/75 (12.0%)	9/45 (20.0%)	4/7 (57.1%)	0.013 *
Dialysis	4/22 (18.2%)	3/9 (33.3%)	0/9 (0.0%)	1/4 (18.2%)	0.202
Previous stroke/TIA	20/127 (15.7%)	9/75 (12.0%)	10/45 (22.2%)	1/7 (14.3%)	0.316
Alcohol abuse	15/127 (11.8%)	10/75 (13.3%)	4/45 (8.9%)	1/7 (14.3%)	0.798
Hypo/hyperthyroidism	14/127 (11.0%)	9/75 (12.0%)	3/45 (6.7%)	2/7 (28.6%)	0.192
Long corticotherapy	11/127 (8.7%)	6/75 (8.0%)	4/45 (8.9%)	1/7 (14.3%)	0.652
Chronic autoimmune disease	9/127 (7.1%)	7/75 (9.3%)	2/45 (4.4%)	0/7 (0.0%)	0.700
Intravenous drug dependency	8/127 (6.3%)	8/75 (10.7%)	0/45 (0.0%)	0/7 (0.0%)	0.059
Intravenous catheter	8/127 (6.3%)	3/75 (4.0%)	5/45 (11.1%)	0/7 (0.0%)	0.290
Immunosuppressive treatment	5/127 (3.9%)	2/75 (2.7%)	3/45 (6.7%)	0/7 (0.0%)	0.519
Previous haemorrhagic events	4/127 (3.1%)	0/75 (0.0%)	3/45 (6.7%)	1/7 (14.3%)	0.021 *
Previous Pulmonary embolism	3/127 (2.4%)	2/75 (2.7%)	1/45 (2.2%)	0/7 (0.0%)	1.000
HIV	1/127 (0.8%)	0/75 (0;0%)	1/45 (2.2%)	0/7 (0.0%)	0.653
Charlson index mean ± SD	3.96 ± 3.04	3.39 ± 2.52	4.51 ± 3.50	6.14 ± 3.29	0.027 *
Antithrombotic treatment at admission	76/127 (59.8%)	32/75 (42.7%)	38/45 (84.4%)	6/7 (85.7%)	<0.001 *
Recent procedures					
Dental procedure	9/127 (7.1%)	7/75 (9.3%)	2/45 (4.4%)	0/7 (0.0%)	0.696
Colonoscopy	7/127 (5.5%)	4/75 (5.3%)	2/45 (4.4%)	1/7 (14.3%)	0.505
Urogenital intervention	5/127 (3.9%)	2/75 (2.7%)	3/45 (6.7%)	0/7 (0.0%)	0.577

CDRIE, cardiac device-related infective endocarditis; COPD, Chronic obstructive pulmonary disease; HIV, Human Immunodeficiency Virus; IE, infective endocarditis; NVE, native valve infective endocarditis; PRVE, prosthetic or repaired valve infective endocarditis; TIA, Transient ischemic attack. A value of *p* < 0.050 was considered significant. (*) denotes significant *p*-values.

**Table 2 jcm-13-01371-t002:** In-hospital mortality of Belgian infective endocarditis patients.

	Total	NVE	PRVE	CDRIE	*p*-Value
N	127	75	45	7	
Death	20/127 (15.7%)	8/75 (10.7%)	9/45 (20.0%)	3/7 (42.9%)	0.051
Cause of death					
Cardiovascular	3/20 (15%)	0/8 (0.0%)	3/9 (33.3%)	0/3 (0.0%)	0.338
Non cardiovascular	12/20 (60.0%)	5/8 (62.5%)	5/9 (55.6%)	2/3 (66.7%)	
Cardiovascular + non cardiovascular	5/20 (25.0%)	3/8 (37.5%)	1/9 (11.1%)	1/3 (33.3%)	
If cardiovascular:					
Heart failure	6/20 (30.0%)	3/8 (12.5)	3/9 (33.3%)	0/3 (0.0%)	
Arrhythmia	1/20 (5.0%)	0/8 (0.0%)	0/9 (0.0%)	1/3 (33.3%)	
Cerebral embolism	1/20 (5.0%)	0/8 (0.0%)	1/9 (11.1%)	0/3 (0.0%)	
If non cardiovascular:					
Sepsis	14/20 (70.0%)	8/8 (100.0%)	4/9 (44.4%)	2/3 (66.6%)	
Neoplasia	1/20 (5.0%)	0/8 (0.0%)	1/9 (11.1%)	0/3 (0.0%)	

CDRIE, cardiac device-related infective endocarditis; NVE, native valve infective endocarditis; PRVE, prosthetic or repaired valve infective endocarditis. A value of *p* < 0.050 was considered significant.

**Table 3 jcm-13-01371-t003:** Multivariate Cox regression analysis for in-hospital all-cause mortality in Belgian infective endocarditis patients.

	Hazard Ratio	95% CI	*p*-Value
Surgical Indication—not performed	19.54	[1.91–200.17]	0.012 *
*S. aureus*	2.99	[1.07–8.33]	0.036

A value of *p* < 0.050 was considered significant. (*) denotes significant *p*-values.

## Data Availability

The data presented in this study are available on request from the corresponding author.

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
