# Peer review of "Infective Endocarditis in Belgium: Prospective Data in Adults from the ESC EORP European Endocarditis Registry"

_jcm, 2024, doi:10.3390/jcm13051371_

Round 1

Reviewer 1 Report

Comments and Suggestions for Authors

Dear Authors,

Congratulations on drafting this relevant manuscript highlighting the characteristics and details of IE in Belgium. My comments are as below

1. The dataset is taken from 1 January 2016 to 31 March 2018, and Follow-up lasted until April 2019. Is it possible to include more patients by extracting data from the last 5 years as well to increase the power as despite using a registry and 6 hospitals, the cohort is relatively small.

2. Can consider including these details in supplementary materials for ease of reading:
- detailed methodology of the ESC EORP EURO-ENDO registry
- diagnosis of definite IE (or possible IE, but considered and treated as IE) 65 based on the ESC 2015 IE guidelines

3. Mention exclusion criteria as well

4. Results, section 3.1 "Seventy-six (56.8%) patients were diagnosed 102 with native valve IE (NVE, group 1)"
This is mentioned in the text, but the table states 75 patients under this heading. Please correct similar errors elsewhere as well if they have been overlooked.

5. For the diagnosis of IE, use the reference of 2023 ESC guidelines: https://academic.oup.com/eurheartj/article/44/39/3948/7243107?login=false#429636718

6. I am finding it difficult to understand supplementary table 3. Can the "effect column" be merged with the 1st column? For example, "indication of surgery" row does not give away a clear meaning.

7. Preferable to write "first" "second" instead of using numbers in the first paragraph of the discussion.

8. The most frequently identified microorganism remains S. aureus, followed by S. viridans and Enterococci --> This would depend on the type of valve so it might not be correct to generalize it. It would be more appropriate if most common organisms on native vs prosthetic valve are mentioned.

9. Other studies from the US and Asian countries should also be compared to identify and highlight the similarities and differences in epidemiology and outcome: PMID: 35614176, PMID: 34861981.
I have included 2 PMIDs here but authors can look into further similar studies.

10. Is it possible to get information of how many of the S. aureus were actually MRSA since that is becoming more common and difficult to treat.

11. "The added value of 18 FDG PET" under discussion section 4.2 can be edited based on the 2023 ESC guidelines which state: 

  • Valvular, perivalvular/periprosthetic and foreign material anatomic and metabolic lesions characteristic of IE detected by any of the following imaging techniques:
    • Echocardiography (TTE and TOE).

    • Cardiac CT.

    • [18F]-FDG-PET/CT(A).

    • WBC SPECT/CT.

Reviewer 2 Report

Comments and Suggestions for Authors

I have thoroughly reviewed the manuscript titled "Infective Endocarditis in Belgium: Prospective Data from the ESC EORP European Endocarditis Registry" by Roosens et al. and appreciate the comprehensive insights it provides into the epidemiology of infective endocarditis in Belgium.

Despite being based on data from five years ago, the manuscript appears robust and aligns with existing literature. However, I have identified some areas for improvement and have provided detailed feedback below:

# The manuscript lacked the most commonly involved valve as it is known elsewhere that the mitral valve is the most commonly affected one. Whenever this is unavailable, this should be mentioned.

# Title:

Capitalize "ESC EORP."

Specify the studied group as "adult" in the title.

# Abstract:

Include tests of statistical analysis in the abstract.

Ensure the conclusion reflects the current results.

# Keywords:

Consider adding the country for better contextualization.

# Introduction:

The introduction is comprehensive and up-to-date but lacks context regarding current global and national epidemiological data. 

Include information on the current positioning of ESC EORP.

# Materials and Methods:

The authors referred to a previous publication for the detailed methodology but they need to describe the design clearly as it is written in the title

Clearly describe the study design as indicated in the title.

Define exclusion criteria.

Clearly define criteria for definite and possible infective endocarditis.

# Results:

Address the discrepancy in the total number of included patients (132 in the text vs. 127 in Table 1).

If the data is well-behaved, reporting mean and SD is sufficient.

Report percentages of males, as infective endocarditis is more common in this group.

Reorganize the history of cardiovascular diseases from most common to least common and when possible apply that to variables in in all tables.

Include an asterisk (*) to denote significant p-values.

Reorganize the % row for chronic autoimmune disease.

Provide more details on microbiological features, particularly for readers with a background in microbiology.

Line 117: Clarify results related to "culture-negative" infective endocarditis as mentioned in the discussion (10%).

Figures were not evaluated and needed to be uploaded in a better quality.

# Discussion:

While insightful, the discussion lacks context beyond the global EURO-ENDO population. Include additional population context for a better explanation of some findings.

Lines 209-211: Provide a brief description of improvements in the diagnostic strategy. 

# The conclusion also needs to reflect the current findings.

In conclusion, the manuscript is well-structured and addresses gaps in the understanding of infective endocarditis in the Belgian population. Addressing the aforementioned comments through a second round of revision would further enhance the manuscript's quality.

Comments on the Quality of English Language

Ok
